

# Changes in satellite retrievals of atmospheric composition over
# eastern China during the 2020 COVID-19 lockdowns
Robert D. Field[1,2], Jonathan E. Hickman[1], Igor V. Geogdzhayev[1,2], Kostas Tsigaridis[1,3], Susanne E. Bauer[1]
[1]NASA Goddard Institute for Space Studies, 2880 Broadway, New York, NY, USA, 10025
[2]Dept. of Applied Physics and Applied Mathematics, Columbia University 2880 Broadway, New York, NY,
USA, 10025
[3]Center for Climate Systems Research, Columbia University, 2880 Broadway, New York, NY, USA, 10025
*Correspondence to*: Robert D. Field (robert.field@columbia.edu)
**Abstract.** We examined daily Level-3 satellite retrievals of AIRS CO, OMI $SO_2$ and $NO_2$, and MODIS AOD
over eastern China to understand how COVID-19 lockdowns affected atmospheric composition, taking into
account trends that have occurred since 2005. Over central east China during the January 23 - April 8 lockdown
window, CO in 2020 was 12% lower than the 2005-2019 mean, but only 2% lower than what would be expected
given the decreasing CO trend over that period. Similarly for AOD, 2020 was 30% lower than the 2011-2019
mean, but not distinct from what would be expected from the trend. $NO_2$ in 2020 was 43% lower than the 2011-
2019 mean, but only 17% lower than what would be expected given the trend over that period. Over southern
China, 2020 $NO_2$ was not significantly different from anticipated, and CO and AOD were significantly higher that
what would be expected, which we suggest was partly because of an active fire season in neighbouring countries.
Over east central and southern China, $SO_2$ was higher than expected, but the magnitude depended strongly on
how daily regional values were calculated from individual retrievals. Future work over China, or other regions,
needs to take these trends into account in order to separate the effects of COVID-19 on air quality from recent
trends, or from variability in other sources.
## 1 Introduction
In an effort to control the spread of COVID-19, the Chinese government implemented a range of restrictions on
movement. These led to reductions in industrial and other work related and personal activities starting January
23, 2020 in Wuhan, Hubei province, then extending to other cities and regions in the days that followed. On April
8, 2020, Wuhan was the last city to re-open after a complete lockdown that prevented most people from leaving
their homes. These measures have been linked to changes in air quality. A network of surface monitoring stations
in northern China observed 35% decreases in $PM_{2.5}$ and 60% decreases in $NO_2$ concentrations during January 29
through February 29, as compared to the preceding three weeks; CO and $SO_2$ also declined (Shi and Brasseur,
2020). In and around Wuhan, decreases of $NO_2$ and $PM_{2.5}$ were similar to regional changes, but there was a slight
increase in $SO_2$ concentrations (Shi and Brasseur, 2020). Observations by the Tropospheric Monitoring Instrument
(TROPOMI) showed large decreases in tropospheric $NO_2$ column densities over Chinese cities, on the order of
40% for February 11 to March 24 2020 compared to the same period in 2019, ranging from roughly 25% for cities
not affected by lockdown to 60% for Wuhan and Xi'an (Bauwens et al., 2020). Prospective simulations suggested
that meteorology may limit the effect of reduced emissions on $PM_{2.5}$ concentrations, with Chinese cities
experiencing less than 20% reductions (Wang et al., 2020).






The goal of our study was to consider these changes against pollution trends in China using NASA Earth
Observing System data by combining several products to give a holistic view covering several emission sectors
that are responsible for the observed changes. Over the last 2 to 3 decades, air pollution in China appears to have
followed the pattern described by the Environmental Kuznets Curve (Selden and Song, 1994). This framework
describes a relationship in which economic growth is initially accompanied by an increase in air pollution, when
poverty remains widespread. But as growth continues, air pollution is expected to level off and decline as a
consequence of changes in social awareness of environmental degradation and the economic, political, and
technological capacity to limit it (Sarkodie and Strezov, 2019;Selden and Song, 1994).

Bottom-up and top-down assessments of air pollutant emissions and concentrations suggest that China has
followed this pattern during the era of satellite monitoring of atmospheric composition, with concentrations of
$SO_2$, $NO_2$, CO, and aerosol optical depth (AOD) mostly exhibiting marked and steady declines over the last
decade. In the case of $NO_2$, multi-instrument analyses, which extend the observational record beyond the lifetime
of a single instrument, depict a consistent regional picture of $NO_2$ trends in China since 1996 (Geddes et al.,
2016;Georgoulias et al., 2019;Wang and Wang, 2020;Xu et al., 2020). Column totals show an increasing trend
during the first part of the satellite record, but this trend is reversed sometime between 2010 and 2014 (Georgoulias
et al., 2019;Krotkov et al., 2016;Lin et al., 2019;Xu et al., 2020;Si et al., 2019;Shah et al., 2020). The trend reversal
has been attributed to a combination of emission control measures (Zheng et al., 2018a) and variations in economic
growth (Krotkov et al., 2016).

Bottom-up estimates suggest that $SO_2$ emissions peaked earlier, with declines starting around 2005, primarily as
a result of power and industrial pollution control measures as well as the elimination of small industrial boilers
(Sun et al., 2018;Zheng et al., 2018b). An earlier peak in $SO_2$ emissions is consistent with observations by multiple
satellite instruments, which revealed declines in $SO_2$ column densities since 2005 (Fioletov et al., 2016;Krotkov
et al., 2016;Wang and Wang, 2020;Zhang et al., 2017;Si et al., 2019).

AOD retrievals from the Along Track Scanning Radiometer instruments show a steady increase over southeastern
China from 1995 to 2005 (Sogacheva et al., 2020), and declines thereafter in the MODIS AOD (He et al., 2019).
The AOD peak has been argued to match either the ~2011 peak in $NO_2$ (Zheng et al., 2018b;Xie et al., 2019), the
~2005 peak of $SO_2$, with more rapid decreases in AOD after 2011 (Lin et al., 2018), or to have occurred in between
(Ma et al., 2016). The recent decrease in AOD is also seen in VIIRS retrievals (Sogacheva et al., 2020). Most
mitigation of direct $PM_{2.5}$ emissions since 2010 was by industry, with residential emissions also decreasing
substantially (Zheng et al., 2018b). The decline in $SO_2$ emissions also exerted an important influence, with the
sulfate concentration of $PM_{2.5}$ decreased substantially between 2013 and 2017 (Shao et al., 2018), reflecting the
negative trend in $SO_2$ emissions.

The peak in concentrations of CO, which has an atmospheric lifetime ranging from weeks to months, is less easily
identified. Some studies suggest that trends have been negative potentially throughout the 21st century (Han et al.,
2018;Strode et al., 2016;Wang et al., 2018;Yumimoto et al., 2014;Zheng et al., 2018a), but others suggest that





emissions and/or column densities were increasing or flat during at least the first decade of the century (Sun et
al., 2018;Zhao et al., 2013;Zhao et al., 2012). The negative trend has been attributed largely to reductions in
emissions from industrial activity, as well as from residential and transportation sectors (Zheng et al.,
2018a;Zheng et al., 2018b).

In addition to these long-term trends, a number of air pollutants also exhibit strong seasonal variation in China.
Anthropogenic emissions of CO, $SO_2$, and $PM_{2.5}$ are highest in winter, reflecting large variation in emissions from
the residential sector and, in the case of CO, increased emissions associated with cold-start processes in the
transportation sector (Li et al., 2017). Outflow of CO and AOD has a spring maximum, resulting from transport
of pollution, dust, and boreal biomass burning emissions (Han et al., 2018;Luan and Jaegle, 2013).

Changes in pollution over China have also come from short-term interventions. To improve air quality for the
2008 summer Olympics—a time when emissions in China were high and still increasing—the Chinese
government imposed a series of strict emissions control measures from July through September 21, 2008, which
were qualitatively similar to the emissions reductions expected to have accompanied the COVID-19 lockdown
(UNEP, 2009). As a result, $NO_2$ concentrations over Beijing were estimated to have declined by between 40%
and 60% based on satellite observations, with substantial but smaller reductions in surrounding cities often on the
order of 20% to 30% compared to previous years (Mijling et al., 2009;Witte et al., 2009). Regional reductions of
$SO_2$ and CO during the months of the games were estimated to be 13% and 19%, respectively (Witte et al., 2009).
These results are broadly consistent with on-road observations (Wang et al., 2009), but larger than some surface
observations comparing concentrations before and after the emission control measures were implemented (Wang
et al., 2010).

The COVID-related lockdowns provide a similar natural experiment to the 2008 Beijing Olympics but on the
other side of the Kuznets curve. The fact that the lockdowns occurred during years of decreasing air pollution
needs to be taken into account in attributing changes in atmospheric composition to COVID-19 lockdowns,
independent of the long-term trend. Following Chen et al.'s (2020) analysis of air quality improvements on
mortality which controlled for changes in air quality since 2016, in this study we determine whether changes in
2020 in satellite retrievals of CO, $SO_2$, $NO_2$ and AOD departed significantly from the expected declines associated
with the long-term decreases in concentrations resulting from  pollution controls and technological change.
**2 Data and methods**
We used daily Level-3 (L3) retrievals from four different instruments on three different NASA Earth Observing
System satellites. The Atmospheric Infrared Sounder (AIRS) instrument aboard NASA's Aqua satellite is a 2300-
channel infrared grating spectrometer in a sun-synchronous orbit with northward equator crossing time of 1:30
PM. AIRS carbon monoxide (CO) profiles are retrieved with horizontal resolution of 45 km at nadir, in a swath
of width 30 fields-of-view or about 1600 km. The retrieval uses a cloud-clearing methodology providing CO with
sensitivity that peaks around 500 hPa, with ~0.8-1.2 degrees-of-freedom-of-signal for 50-70% of scenes. More
sampling and higher information content is obtained in clear scenes (Warner et al., 2013). We used the daily
version 6 (AIRS3STD.006) product.




The Ozone Monitoring Instrument (OMI) aboard NASA's Aura satellite was launched in July 2004, and has a
local equator-crossing time of roughly 13:45. OMI is a nadir-viewing spectrometer, which measures solar
backscatter in the UV-visible range (Krotkov, 2013). We used NASA's L3 tropospheric $NO_2$ column density
Standard Product v3 (OMNO2d_003), and the OMI Principle Components Analysis Planetary Boundary Layer
(PBL) $SO_2$ product (OMSO2e_003), which grid retrievals to 0.25º resolution (Krotkov et al., 2017;Li et al., 2013).
Both products are cloud-screened; only pixels that are at least 70% cloud-free are included in the $NO_2$ product,
and those that are at least 80% cloud-free are included in the $SO_2$ product. The $NO_2$ product relies on air mass
factors (AMFs) calculated with the assistance of an atmospheric chemical transport model and are sensitive to
model representations of emission, chemistry, and transport data. Instead of AMFs, the $SO_2$ product uses
spectrally-dependent $SO_2$ Jacobians, but can be interpreted as having a fixed AMF that is representative of
summertime conditions. We applied basic transient $SO_2$ plume filtering, excluding retrievals with $SO_2 > 15$ DU
(Wang and Wang, 2020).

Because our trend analysis uses a seasonal mean as the response variable, we assume that random errors cancel
out, leaving only systematic errors, which do not contribute to uncertainty in the trend analysis. Systematic errors
in the OMI $NO_2$ product (associated with AMFs and tropospheric vertical column contents) have an uncertainty
of 20% (McLinden et al., 2014). The OMI $NO_2$ products use an implicit aerosol correction to account for the
optical effects of aerosols, but retrievals can be biased when aerosol loading is extreme (Castellanos et al., 2015).
Under these conditions, the OMI $NO_2$ retrieval is biased low by roughly 20 to 40% (Chimot et al., 2016). Note
that any aerosol-related error would have the potential effect of underestimating the magnitude of decreases in
$NO_2$ column densities when comparing 2020 to previous years. Additional bias in the $NO_2$ product may be
introduced due to the reliance on nearly cloud-free pixels, in which greater sunlight may induce higher
photochemical rates. For example, the current $NO_2$ product is biased roughly 30% low over the Canadian oil sands
(McLinden et al., 2014). The level-2 OMI- $NO_2$ product has been validated against in situ and surface-based
observations showing good agreement (Lamsal et al., 2014). The use of fixed Jacobians in the $SO_2$ product
introduces systematic errors of 50 to 100% for cloud-free observations (Krotkov et al., 2016).

Starting in 2007, the quality of level 1B radiance data for some OMI viewing directions has been affected, known
as the row anomaly. The L3 products used here exclude all pixels affected by the row anomaly from each
observation, but the locations of the row anomaly pixels were dynamic between 2007 and 2011, which could
affect any comparisons including those years. Since 2011, the pixels affected by the row anomaly problem are the
same, so comparisons for data only since 2011 are not affected by changes in the row anomaly.

Moderate Resolution Imaging Spectroradiometer (MODIS) sensors observe the Earth from polar orbit, from Terra
satellite since 2000 and from Aqua since mid 2002. In this study we use MODIS-derived AOD at 550nm obtained
by merging Dark Target and Deep Blue retrievals (Sayer et al., 2014). Specifically, we use the
Deep_Blue_Aerosol_Optical_Depth_550_Land_Mean    field    over    land    and    the    over    ocean
AOD_550_Dark_Target_Deep_Blue_Combined_Mean the from Collection 6.1 L3 Gridded products MYD08 and
MOD08 (Hubanks et al., 2019), though very few retrievals over ocean are included in our analysis. L3 values are



computed on $1° \times 1°$ spatial grid from L2 AOD products with resolution of 10x10 km. Over land 66% of MODIS-
retrieved Dark-target AOD values were shown to be ±0.05±.15*AOD AERONET-observed values, with high
correlation (R = 0.9) (Levy et al., 2010). Around 78% of the Deep Blue retrievals are within the expected error
range of ±0.05±0.20*AOD (Sayer et al., 2013). MODIS AOD data have been extensively used by the modeling
and remote sensing scientific communities and inter-compared with a wide range of satellite AOD products (see
Schutgens et al. (2020) and references therein).

We analyzed these retrievals over two large regions (Fig. 1). Central east China was comprised of Shaanxi, Hubei,
Anhui, Jiangsu, Shanxi, Henan, Hebei, Shandong, Beijing, and Tianjin provinces. Southern China was comprised
of Guizhou, Guangxi, Hunan, Jiangxi, Guangdong, Fujian and Zhejiang provinces. Daily mean quantities were
calculated across all valid retrievals falling within the provinces comprising the regions. For the OMI $NO_2$
columns, individual retrievals were weighted by the L3 'Weight' field, which is proportional to the fraction of the
grid cell with higher-quality retrievals. We also calculated the daily value from the median of all retrievals, to
understand whether individual high values (mainly $SO_2$) had any effect on the significance of trends or differences
between 2020 and different background periods. Monthly averages were calculated from the daily regional
averages, with each day weighted in the monthly average by the number of valid retrievals so as to not
overrepresent days with little satellite coverage or significant cloud cover. The monthly data were used to visually
identify COVID-19 related changes against background seasonality and trends since 2005.

We examined the difference in the distribution of daily data during the 2020 January 23 to April 8 lockdown
period to the same period during previous years since 2005. We compared 2020 to 2019, to background periods
ending in 2019 over which trends were consistent, and to the expected value for 2020 estimated from these trends.
We tested the significance of these differences using bootstrap resampling (Efron and Gong, 1983) with a
resampling size of 2000. Given the uncertainty and uneven nature of trends over different parts of China from
previous studies, we identified the start of existing long-term trends for each species by conducting linear
regressions of the change in the four quantities over time for possible start years of 2005 to 2015. Each trend was
estimated from the start year in this range until 2019. We selected the start year for the most significant trend and
used that trend for comparisons to 2020 data.

We also considered how the analysis depended on the how the lockdown period was defined. Emissions and
pollution can decrease during the Chinese New Year holidays (Chen et al., 2020), which started as early as January
23 in 2012 and as late as February 19 in 2015, complicating COVID-19 related analyses of atmospheric
composition over China (Bauwens et al., 2020;Chen et al., 2020). The timing and extent of lockdowns also varied
between provinces and we assume that 'slowdowns' could have happened before or after stricter, official
lockdowns. For example, ground and air transportation remaining below lockdown levels nationally at least
through April 14, 2020 (International Energy Agency, 2020). Excluding the holiday period from all years is a
straightforward approach to excluding any New Year holiday effects but will exclude simultaneous lockdown
effects during the initial, and presumably most strict, stages of the lockdown. Rather than specifying different
combinations of New Year holiday period and provincial-level lockdown timing, we used January 23-April 8 as
our baseline period (which will include all holiday periods since 2005), but examined the sensitivity of the


statistics to the length of the lockdown period, namely a longer lockdown period beginning one week earlier and
one week later, and a shorter lockdown beginning one week later and ending one week earlier. In interpreting the
data, we put more confidence in 2020 differences that were insensitive to these choices.
**3 Results**
**3.1 Regional patterns and seasonality**
Figure 2 shows the 2020 –2019 differences over China during the January 23-April 8 lockdown period for the
four satellite-retrieved quantities. There were decreases of 5-10 ppbv in AIRS CO over central east China (Fig.
2a) and increases of 20-25 ppbv over southern China in 2020 compared to 2019. The increase in southern China
is adjacent to a stronger positive CO anomaly over the upper Mekong regions of Myanmar, Thailand and Laos.
There were no coherent regional changes in OMI $SO_2$ (Fig. 2b), but rather smaller localized difference of either
sign. There were decreases in $NO_2$ (Fig. 2c) across central east China exceeding $8\times10^{15}$ molec cm$^{-2}$ coincident
with the weaker decrease in CO. Over southern China, there were comparable differences over Guangdong
province, with smaller differences elsewhere. There was a decrease in MODIS AOD (Fig. 2d) in central-east
China coincident with the decreases in CO and $NO_2$, but smaller in magnitude. There was a region of higher AOD
in and northeast of the upper Mekong region coincident with the CO increase, both presumably because of biomass
burning.

To put the 2020/2019 difference maps in a longer-term and seasonal context, Figure 3 shows monthly averages
of the four retrieved quantities over central east China since 2005. There are seasonal CO peaks in March-April,
June and September, with the minima usually in November and December (Figure 3a). There has been a decrease
since 2005 in CO. The seasonal decrease from January to February in 2020 is similar to that which has occurred
occasionally before, but the CO during February and March 2020 was the lowest for that time of the year since
2005. By April, CO had returned to levels typical of 2015-2019. The main features of the $SO_2$ are that it has
decreased since 2005 (Figure 3b), and that early 2020 $SO_2$ was within the range of recent levels. There is a strong
seasonal $NO_2$ cycle (Figure 3c), with a July-August minimum, and December-January peak, which has been
attributed to increased heating needs (Yu et al., 2017;Si et al., 2019) and longer chemical lifetime owing to lower
OH and $RO_2$ (Shah et al., 2020). $NO_2$ has also decreased since 2011, and during most years, there is a departure
from a smooth seasonal cycle in January and February associated with the Chinese New Year holiday period.
January and February 2020 $NO_2$ was considerably lower than previous years, increased during March, and had
recovered to typical, recent levels by April. AOD has consistent seasonal peaks in summer which have been
attributed to hygroscopic growth and agricultural residue burning (Filonchyk et al., 2019), but had less regular
seasonality otherwise, and has decreased since 2011. AOD during February and particularly March of 2020 were
lower than recent years, but against a noisy background.

Figure 4 shows the four retrieved quantities over southern China. There is a springtime maximum in CO (Fig. 4a),
a less regular maximum during September-January, and an annual minimum in July. The range of CO is similar
to central east China. CO over the last 5 years is lower than earlier in the record, and early 2020 CO was higher
than recent years. $SO_2$ (Fig. 4b) is lower than central east China and any seasonal cycle is also hard to identify.



The high June 2011 values are due to the Nabro eruption in Ethiopia (Fromm et al., 2014) which is still apparent
in the time series despite excluding individual $SO_2$ retrievals that are greater than 15 DU. $NO_2$ (Fig. 4c) is lower
than over east central China, but both regions share a similar seasonality. $NO_2$ during January-April 2020 was
slightly lower than 2019. AOD (Fig. 4d) has weak seasonal peaks in October, March and June, has decreased
since 2011, and 2020 fell within the range of 2015-2019.

**3.2 East central China**
Figure 5 shows the distribution of daily CO, $SO_2$, $NO_2$ and AOD for January 23 – April 8 of each year over east
central China. The associated statistics comparing 2020 and 2019 are provided in Table 1, and comparing 2020
with the period over which the trend is consistent in Table 2. The linear trends in each plot start at the year over
which the trend explains most of the variability in the data, which will vary by region and variable. The daily
distribution of AIRS CO (Fig. 5a) is shown by the black box and whisper plots. The variation during January 23
– April 8 of each year is due to weather-related factors and observational error. The mean CO of 133.5 ppbv in
2020 was 3% less than the 2019 mean of 137.9 ppbv, and 12% less than 2005-2019 mean of 150.9 ppbv. The
2020 difference from 2019 is only marginally significant, with a 95% confidence interval (-6% - 0%) close to
spanning 0. The 2020 difference from the 2005-2019 background is significant (-14% - -9%), but during this
period CO declined by -1.8 ppbv $yr^{-1}$, indicated by the red points. This overall decrease includes periods where
CO may have increased, for example from 2010-2012, 2016 and slightly in 2019. Based on this trend, the expected
value for 2020 was 136.8 ppbv (shown in blue). The observed 2020 mean was 2% less than expected, but because
the 95% confidence intervals (-5% - 1%) span 0, this difference is not considered to be significant.

OMI $SO_2$ (Fig. 5b) fluctuated over 2005 to 2011 and declined steadily afterward by -.056 DU $yr^{-1}$ from 2012-
2019 over east central China. This trend explained 32% of the variation in the data over this period, during which
overall variation declined, becoming narrower to a degree not seen in the CO. Declines were steady, although
2019 may have departed upward from this trend. The 2020 mean of 0.057 was 93% higher than the 2019 mean of
0.032, but with a wide 95% confidence interval (16% - 236%). 2020 $SO_2$ was 72% lower than the 2012-2019
mean, and with a narrower (-78% - -65%) confidence interval. The observed 2020 $SO_2$ was 201% higher than the
expected value of -0.06. The change in 2020 $SO_2$ was strongly dependent on whether daily values were calculated
from the mean or median of individual values over the region. When the median of individual retrievals is used,
2020 was only 8.4% higher than predicted from the 2012-2019 trend (Figure S1b). This likely reflects the greater
influence of high individual retrieval values on the daily mean value compared to the median, even after the basic
filtering of transient $SO_2$ plumes.

OMI $NO_2$ (Fig. 5c) increased from 2005 to 2011 and decreased by -0.7x$10^{15}$ molec $cm^{-2}$ $yr^{-1}$ from 2011-2020. The
2020 mean $NO_2$ of 6.5x$10^{15}$ molec $cm^{-2}$ was 32% less than the 2019 mean of 9.6 x$10^{15}$ molec $cm^{-2}$, and 43% less
than the 2011-2019 mean of 11.3 x$10^{15}$ molec $cm^{-2}$. The pronounced regional difference between 2020 and 2019
(Fig. 2c) in part likely reflects an upward departure in 2019 from the overall trend since 2011. The observed 2020
mean was 17% less than the expected value of 7.8 x$10^{15}$ molec $cm^{-2}$, with a wide but negative 95% confidence
interval (-28% - -5%), suggesting that 2020 $NO_2$ was significantly lower than would be expected from the trend.




MODIS AOD (Fig. 5d) was flat or slightly increasing from 2005 to 2011 and subsequently changed by -0.03 yr⁻
¹. The 2020 mean AOD of 0.41 was 15% less than the 2019 mean of 0.48 and 30% less than the 2011-2019
average of 0.58. The observed 2020 mean was 2% higher than the predicted value of 0.40, but with a wide (-15%
- 20%) confidence interval spanning 0, suggesting 2020 was not significantly different from expected.

In evaluating the 2020 changes, the background period was defined by the period during which the trend was
strongest, using the $r^2$ value of the trend. This is a reasonable but ad-hoc way of defining a period with consistent
increasing or decreasing trends. Figure 6 shows how the trends and differences between observed and predicted
2020 means depended on the year chosen as the start of the period over which the trend is estimated. AIRS CO
(Fig. 6a) showed uneven changes in the trend (red line) with starting year, and more uncertainty in the trend (red
shading) for later years due to fewer data used for the estimate, but for all years was significantly negative. The
difference between observed 2020 mean and the value predicted from the trend (magenta line) varied inversely
with the trend and was always negative, but, except for 2009, had 95% confidence intervals (magenta shading)
spanning 0, and therefore were not considered significant. The $SO_2$ trends (Fig. 6b) were all significantly negative.
For trends starting in 2007 and after, the observed 2020 mean was significantly higher than predicted, but these
differences were not consistently significant when daily values were calculated from the median of individual
retrievals (Figure S2b). Earlier starting years produce weaker overall trends in $NO_2$ (Fig. 6c) because of the $NO_2$
increase until 2011, but observed 2020 $NO_2$ was significantly less than predicted regardless of the starting year.
Note that analyses of $SO_2$ and $NO_2$ that include years prior to 2012 may be affected by changes in observation
sample size due to changes in the OMI row anomaly. For AOD (Fig. 6d), there was no significant difference
between the observed and predicted 2020 mean for periods beginning in 2009 and later, when the trends were
strongest.
**3.3 Southern China**
Figure 7 shows the distribution of daily CO, $SO_2$, $NO_2$ and AOD for January 23-April 8 of each year over southern
China, along with linear trends. The associated statistics comparing 2020 and 2019 are provided in Table 3, and
comparing 2020 with the period over which the trend is consistent in Table 4. For AIRS CO (Fig. 7a), the strongest
trend started in 2007 and was -1.8 ppbv yr⁻¹ through 2019. CO in 2020 was 144.8 ppbv, 13% higher than the 2019
mean of 128.4 ppbv which can be seen in an upward shift in the distribution of the box plot, and nearly identical
to the 2007-2019 background period mean of 144.9 ppbv. 2020 CO was 10% higher than predicted from the 2007-
2019 trend, and with 95% confidence interval (5% - 15%) not spanning 0.

OMI $SO_2$ (Fig. 7b) changed by -0.012 DU yr⁻¹ beginning in 2007, which is driven by fewer high individual $SO_2$
values in later years, as in east central China. The 2020 mean of 0.003 DU was 115% higher than the 2019 mean
of -0.023 DU with a wide but positive 95% confidence interval (32% - 215%), and 95% less than the 2007-2019
mean of 0.058 DU. The observed 2020 mean was 109% higher than the predicted value of -0.034 DU, with a wide
but positive 95% confidence interval (54% - 164%).





OMI $NO_2$ (Fig. 7c) changed by -0.3 $\times 10^{15}$ molec $cm^{-2}$ $yr^{-1}$ beginning in 2011. The 2020 mean of 3.3x$10^{15}$ molec
$cm^{-2}$ was 22% less than 2019 and 32% less than 2011-2019, with both differences significant. The 2020 mean was
7% less than the predicted value of 3.6 x$10^{15}$ molec $cm^{-2}$, but with a wide 95% confidence interval (-20% - 8%)
spanning 0.

MODIS AOD (Fig. 7d) changed by -0.04 $yr^{-1}$ between 2011 and 2019. The 2020 mean AOD of 0.38 was 12%
higher than the 2019 mean of 0.34, but with a 95% confidence interval (-6% - 32%) spanning 0. 2020 AOD was
17% less than the 2011-2019 mean of 0.46, but 39% higher than predicted from the trend over this period, and
with a wide but positive 95% confidence interval (16% - 65%).

Figure 8 shows the dependence of the trends and 2020 differences from background period to the starting year
over southern China. AIRS CO (Fig. 8a) had a significant decreasing trend for all starting years, and regardless
of the start year, 2020 was significantly higher than predicted from the trend. $SO_2$ trends were also negative, and
varied similarly to the CO. The 2020 $SO_2$ was higher than the background period, but with marginal significance,
given that the confidence intervals spanned 0 for later starting years, and because the differences were not
significant when daily values were calculated from the median $SO_2$ of individual retrievals for trends starting in
2008 (Figure S3b) or any other year (Figure S4b). $NO_2$ trends (Fig. 8c) were more strongly decreasing for periods
beginning between 2009 and 2012 and were flat or positive otherwise. The 2020 $NO_2$ mean was significantly
lower than predicted, except for when the trend was estimated beginning in 2011, when it was the strongest. The
AOD trends varied similarly to the $NO_2$ but were significantly negative for all start years. The 2020 mean was
significantly higher than predicted for all starting between 2010 and 2015.

For both regions and all quantities, the differences between observed and predicted values for 2020 were
insensitive to a longer or shorter lockdown period, or to whether the bootstrap resampling was weighted by the
number of valid retrievals each day.
**4 Discussion and conclusions**
The degree to which the COVID-19 lockdowns in China resulted in changes in atmospheric composition depends
strongly on whether existing trends are taken into account, and only in certain cases could be considered
significant. For AIRS CO over central east China, the 2020 mean was 12% less than that over 2005-2019, but
only 2% less than what would be expected given the steady decreases over that period, and this 2% was not
significant given the variability of the daily data. Similarly for MODIS AOD, the 2020 mean was 30% less than
over 2011-2019, but no different than what would be expected from trends. $SO_2$ in 2020 was 72% less than over
2012-2019 but was 201% higher than what would be expected from trends. Daily $SO_2$ calculated from the mean
of individual retrievals are sensitive to outlying $SO_2$ values from transient plumes, and when daily $SO_2$ was
calculated instead from the median across individual retrievals, 2020 $SO_2$ was only 8% higher than what would
be expected, but still significantly.

OMI $NO_2$ in 2020 over central east China was 43% less than over 2011-2019, but only 17% less than what would
be expected from trends. This difference was statistically significant but does suggest that more than half of the



reductions in $NO_2$ in 2020 could be expected independent of COVID-19 lockdowns. For reference, Bauwens et
al. (2020) reported a ~40% drop in OMI $NO_2$ from 2019 to 2020 over cities affected by the lockdown using the
QA4ECV retrieval (Boersma et al., 2018), and a ~51% drop in $NO_2$ over the eight cities (Beijing, Jinan, Nanjing,
Qingdao, Tianjin, Wuhan, Xi'an and Zhengzhou) falling within our central east China region. Our analysis cannot
be compared directly because we include non-urban areas and define our lockdown period differently, but we can
say that a large part of the reduction in that study is likely due to background trends, rather than to COVID-19
lockdowns.

The lack of any significant departure from recent trends in CO and AOD over central east China was unexpected,
given its high population density and level of industrial activity. In the case of MODIS AOD, the lack of an
observable lockdown effect was possibly due to contributions from other sources unaffected by COVID-19 related
lockdowns, limitations in the MODIS AOD retrieval under cloudy conditions, climatological variability from
other sources such as mineral dust, and meteorology favorable to secondary aerosol formation which could have
offset lower emissions (Wang et al., 2020). The 2020 increase in $SO_2$ is more difficult to interpret because of the
discrepancies between daily values calculated from the mean or median of individual retrievals, but is broadly
consistent with surface observations that find no significant change in in-situ surface $SO_2$ over Wuhan in the daily
mean, and a slight increase in daytime $SO_2$ possibly associated with increased residential heating and cooking
(Shi and Brasseur, 2020).

Over southern China, retrieved $SO_2$, $NO_2$ and AOD were significantly lower in 2020 compared to recent averages.
$SO_2$ was 95% less than the 2007-2019 mean, but 109% greater than what would be expected from trends. Similarly
to central east China, $SO_2$ was only 5% higher than expected when daily values were calculated from the median
of individual retrievals, rather than the mean. $NO_2$ in 2020 was 32% less than over 2011-2019, but only 7% less
than what would be expected from trends, and this difference was not consistently significant for different trend
periods. The more significant reductions in $NO_2$ in east central China compared to the south is consistent with
Chen et al.'s (2020) detection of a larger 2020 decrease in surface $NO_2$ in Wuhan compared to Shanghai. Retrieved
CO in 2020 was nearly identical to the 2007-2019 mean, but 10% higher than what would be expected given the
decreasing trend over this period. AOD in 2020 was 17% less than over 2011-2019, but 39% higher than what
would be expected from the trend.

The focus of this analysis is on whether satellite retrievals of atmospheric composition over 2020 departed
significantly from different background periods and expected values for 2020 when daily variability and trends
are accounted for, but it is useful at a preliminary stage to speculate as to how different emissions changes could
have contributed to 1) why $NO_2$ was robustly lower in 2020 over east central China compared to CO and AOD,
and 2) why CO and AOD were higher over southern China compared to what would be expected from recent
trends.

To understand why $NO_2$ differences over east central China were more significant than other quantities, Table 5
shows the emissions by sector for a representative set of constituents from the Community Emissions Data System
(CEDS) (Hoesly et al., 2018) over China for 2014, the most recent year available. Other bottom-up emissions



inventories will vary in absolute emissions amounts and their sector contributions, particularly for more recent
periods, but CEDS is the standard available emissions dataset available globally as a baseline for the next IPCC
assessment, in anticipation of assessing 2020 COVID-19 related changes to atmospheric composition in other
regions, and for modeling studies involving a transboundary transport component. Across all species, energy
production, industrial activity, transportation, residential/commercial/other (RCO), and waste disposal constitute
the bulk of the emissions. Based on activity data for the first quarter of 2020, energy demand across China declined
by 7% compared to 2019, and transportation sector activity declined by 50 to 75% in regions with lockdowns in
place (International Energy Agency, 2020). These sectors are direct or indirect sources of numerous pollutants,
including $SO_2$ (the precursor of sulfate aerosol), $NO_x$, CO, and primary anthropogenic aerosols classified broadly
as organic carbon (OC) and black carbon (BC). If we apply the 7% reduction in energy production and mid-point
62.5% reduction to transportation from the IEA, assume a 20% reduction in industrial emissions, 5% reduction in
waste emissions, no change in RCO (with commercial decreases offset by residential increases), this yields a 10%
reduction in BC, 5% reduction in OC, 14% reduction in $SO_2$, 14% reduction in CO and 21% reduction in $NO_2$.
The larger reduction in $NO_2$ relative to other emissions could partly explain why OMI $NO_2$ column density
changes over central east China were stronger than in the other retrievals.
Following Si et al.'s (2019) consideration of biomass burning as a pollution source in China alongside
anthropogenic sources, we considered transboundary smoke transport as a possible reason for the higher 2020 CO
over southern China, guided by higher CO over the Upper Mekong region in 2020 compared to 2019 (Fig. 2a).
Table 6 compares January 23-April 8 AIRS CO over southern China to CO emissions estimates from biomass
burning from the Global Fire Assimilation System (GFAS) (Kaiser et al., 2012) over the upper Mekong region
($17^o$ N to $25^o$ N, $95^o$ E to $105^o$ E) including parts of eastern Myanmar, northern Thailand, and northern Laos. From
2005 to 2020, variation in GFAS CO over this region explained a moderate (32%) amount of variability in AIRS
CO over southern China, suggesting it as is a non-negligible contributor to variation in CO concentration, and a
contributor to higher CO in 2020. This illustrates that, at a minimum, sources such as biomass burning smoke and
dust that are less affected by COVID-19 related measures will complicate attribution studies. To that end,
modeling studies following Wang et al. (2020) will be required to isolate emissions, meteorological and chemical
drivers of changes in atmospheric composition and their effects.
The key implication of our study is that not taking into account past trends in atmospheric composition will lead
to misattribution of changes in air quality to COVID-19 lockdowns. We have approached the issue by comparing
data for 2020 to what would have been expected given recent trends, and by applying a single lockdown period
to two large regions. Other studies over China or elsewhere will inevitably use other approaches that more
explicitly account for seasonality and which relate changes in pollution over smaller areas (e.g. single provinces
or states) to region-specific lockdown measures and timing. Regardless of the approach, however, it is important
to consider recent trends and variability. In places where pollution has decreased, not accounting for recent context
will result in over-attribution of changes in pollution to COVID-19. In places where pollution has increased, such
as parts of South Asia, this will result in under-attribution.



**Code/data availability:** All code will be made available if the article is accepted for final publication. All source
data are publicly available.

**Competing interests:** The authors have no competing interests.

**Author contribution:** All authors conceived of the study. RF, IG and KT conducted the data analysis. RF and
JH prepared the manuscript with contributions from all co-authors.



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



**Tables**

**Table 1. Summary statistics for central east China comparing 2020 and 2019 during January 23 – April 8.**

| Variable | 2020 mean | 2019 mean | 2020 % difference from 2019 |
|---|---|---|---|
| **CO** | 133.5 | 137.9 | -3 |
| (ppbv) | (130.4, | (134.6, | (-6, |
| | 136.8) | 141.2) | 0) |
| | | | |
| **SO$_2$** | 0.057 | 0.032 | 93 |
| (DU) | (0.045, | (0.018, | (16, |
| | 0.070) | 0.046) | 236) |
| | | | |
| **NO$_2$** | 6.5 | 9.6 | -32 |
| ($10^{15}$ | | | |
| molec cm$^{-2}$) | (5.7, | (8.7, | (-42, |
| | 7.2) | 10.5) | -22) |
| | | | |
| **AOD** | 0.41 | 0.48 | -15 |
| | (0.36, | (0.41, | (-30, |
| | 0.46) | 0.55) | 3) |







**Table 2. Summary statistics for central east China during January 23 – April 8 over consistent trend periods. The**
**background start year is that for which estimated trends explained most variability in the daily data through 2019, judging**
**by the coefficient of determination ($r^2$) of the estimated trend. The background means and trends are estimated from the**
**background start year through 2019, and the 2020 prediction is calculated from this trend. Numbers in parentheses are**
**bootstrap-estimated 95% confidence intervals. The trends and percent differences are considered to be significant if their**
**confidence intervals do not span 0.**

| Variable | Background start year | Background mean | 2020 difference from background | trend ($yr^{-1}$) | trend $r^2$ | 2020 predicted from trend | 2020 % difference from trend |
|---|---|---|---|---|---|---|---|
| **CO** | 2005 | 150.9 | -12 | -1.8 | 0.17 | 136.8 | -2 |
| (ppbv) | | (149.8, | (-14, | (-2.0, | (0.13, | (134.8, | (-5, |
| | | 152.1) | -9) | -1.5) | 0.21) | 138.8) | 1) |
| **SO₂** | 2012 | 0.206 | -72 | -0.056 | 0.32 | -0.061 | 201 |
| (DU) | | (0.184, | (-78, | (-0.065, | (0.26, | (-0.090, | (158, |
| | | 0.229) | -65) | -0.048) | 0.38) | -0.033) | 277) |
| **NO₂** | 2011 | 11.3 | -43 | -0.7 | 0.17 | 7.8 | -17 |
| ($10^{15}$ | | | | | | | |
| molec $cm^{-2}$) | | (10.9, | (-49, | (-0.8, | (0.11, | (7.1, | (-28, |
| | | 11.7) | -36) | -0.5) | 0.22) | 8.5) | -5) |
| **AOD** | 2011 | 0.58 | -30 | -0.03 | 0.08 | 0.4 | 2 |
| | | (0.55, | (-39, | (-0.05, | (0.04, | (0.35, | (-15, |
| | | 0.60) | -20) | -0.02) | 0.13) | 0.45) | 20) |




**Table 3. Same as Table 1, but for southern China.**

| Variable | 2020 mean | 2019 mean | 2020 % difference from 2019 |
|---|---|---|---|
| **CO** | 144.8 | 128.4 | 13 |
| (ppbv) | (139.4, | (124.1, | (7, |
|  | 150.3) | 132.6) | 19) |
|  |  |  |  |
| **SO$_2$** | 0.003 | -0.023 | 115 |
| (DU) | (-0.013, | (-0.039, | (32, |
|  | 0.019) | -0.009) | 215) |
|  |  |  |  |
| **NO$_2$** | 3.3 | 4.3 | -22 |
| ($10^{15}$ |  |  |  |
| molec cm$^{-2}$) | (3.0, | (3.9, | (-32, |
|  | 3.6) | 4.7) | -10) |
|  |  |  |  |
| **AOD** | 0.38 | 0.34 | 12 |
|  | (0.33, | (0.30, | (-6, |
|  | 0.43) | 0.39) | 31) |






**Table 4. Same as Table 2, but for southern China.**

| Variable | Background start year | Background mean | 2020 difference from background | trend (yr⁻¹) | trend r² | 2020 predicted from trend | 2020 % difference from trend |
|---|---|---|---|---|---|---|---|
| **CO** | 2007 | 144.9 | 0 | -1.8 | 0.09 | 132.1 | 10 |
| (ppbv) | | (143.3, | (-4, | (-2.2, | (0.05, | (129.5, | (5, |
| | | 146.5) | 4) | -1.4) | 0.13) | 134.8) | 15) |
| **SO₂** | 2007 | 0.058 | -95 | -0.012 | 0.13 | -0.034 | 109 |
| (DU) | | (0.049, | (-123, | (-0.014, | (0.09, | (-0.049, | (54, |
| | | 0.068) | -65) | -0.009) | 0.17) | -0.019) | 164) |
| **NO₂** (10¹⁵ | 2011 | 4.9 | -32 | -0.3 | 0.08 | 3.6 | -7 |
| molec cm⁻²) | | (4.7, | (-39, | (-0.3, | (0.05, | (3.2, | (-20, |
| | | 5.1) | -25) | -0.2) | 0.13) | 3.9) | 8) |
| **AOD** | 2011 | 0.46 | -17 | -0.04 | 0.18 | 0.27 | 39 |
| | | (0.44, | (-27, | (-0.04, | (0.13, | (0.24, | (16, |
| | | 0.48) | -6) | -0.03) | 0.23) | 0.31) | 65) |






**Table 5. 2014 anthropogenic emissions estimates by sector (in %) over China, excluding biomass burning, from the Community Emissions Data System (CEDS) for a representative set of constituents: black carbon (BC), carbon monoxide (CO), ammonia (NH₃), nitrogen oxides (NOₓ), organic carbon (OC) and sulfur dioxide (SO₂). Residential, commercial and other sectors are combined as RCO.**

|  | BC | CO | NH₃ | NOx | OC | SO₂ |
|---|---|---|---|---|---|---|
| Agriculture | 0 | 0 | 61.6 | 1.1 | 0 | 0 |
| Energy | 32.6 | 8 | 0.4 | 38.5 | 28.3 | 29.4 |
| Industrial | 12.7 | 41.8 | 6.5 | 33 | 5.1 | 57.3 |
| Ground transportation | 8.1 | 7.2 | 0.5 | 17.5 | 1.7 | 0.3 |
| RCO | 38.1 | 36.7 | 5.2 | 4.2 | 38.4 | 12.5 |
| Solvents | 0 | 0 | 0 | 0 | 0 | 0 |
| Waste | 8.5 | 6.3 | 25.8 | 5.2 | 26.5 | 0.4 |
| Shipping | 0 | 0 | 0 | 0.2 | 0 | 0.1 |
| Aircraft | 0 | 0 | 0 | 0.2 | 0 | 0 |





**Table 6. Bottom up biomass Global Fire Assimilation System (Kaiser et al., 2012) burning CO emissions estimates from the**
**Upper Mekong region (17º N to 24º N, 95º E to 105º E) and AIRS CO over southern China from January 23 to April 8, for**
**2005-2020.**

| Year | GFAS CO Upper Mekong (KT) | AIRS CO southern China 500 hPa (ppbv) |
|---|---|---|
| 2005 | 7977 | 157 |
| 2006 | 8905 | 146 |
| 2007 | 15734 | 165 |
| 2008 | 4542 | 153 |
| 2009 | 9990 | 140 |
| 2010 | 14176 | 149 |
| 2011 | 3591 | 147 |
| 2012 | 11320 | 153 |
| 2013 | 8684 | 145 |
| 2014 | 8722 | 142 |
| 2015 | 8084 | 143 |
| 2016 | 9642 | 149 |
| 2017 | 3736 | 131 |
| 2018 | 3179 | 139 |
| 2019 | 6309 | 128 |
| 2020 | 7871 | 145 |






**Figures**

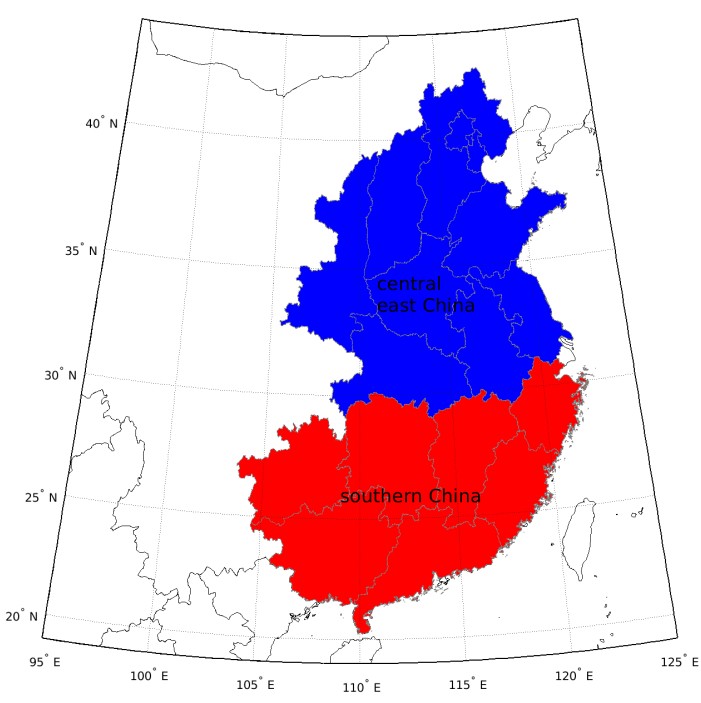


**Figure 1. Groupings of provinces for central east China and southern China.**

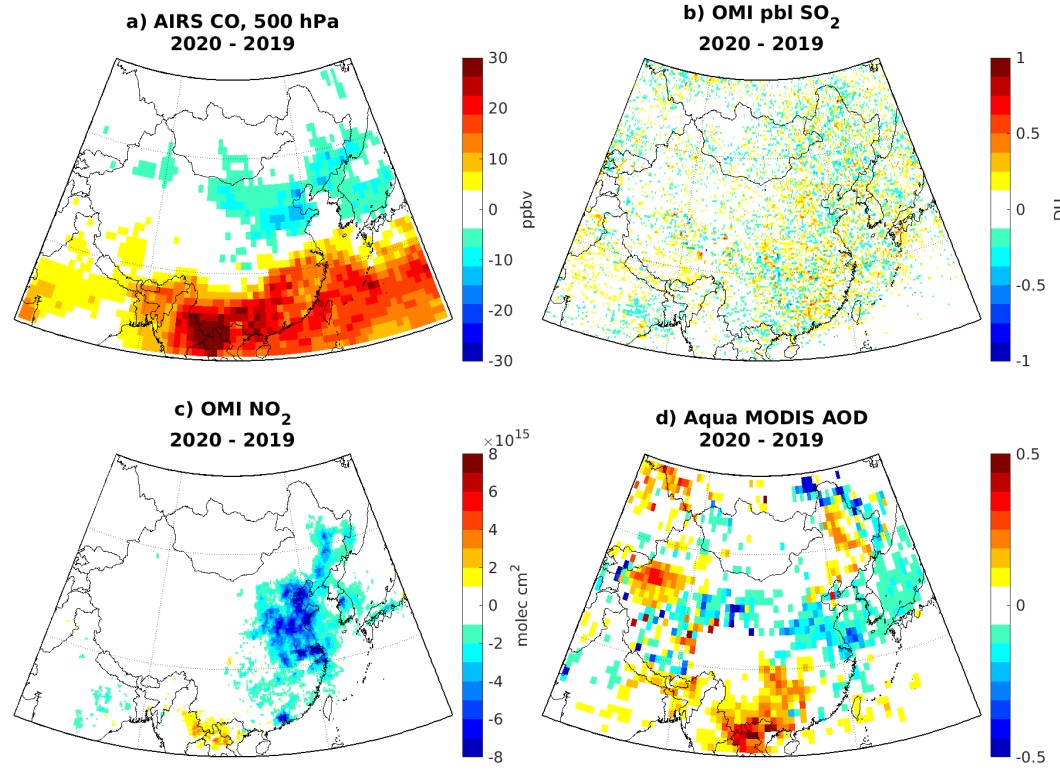


**Figure 2. 2020-2019 differences during January 23 to April 8 over China in a) AIRS carbon monoxide (CO) at 500 hPa, b) OMI PBL sulfur dioxide (SO₂), c) OMI tropospheric nitrogen dioxide (NO₂) and d) Aqua MODIS aerosol optical depth (AOD).**





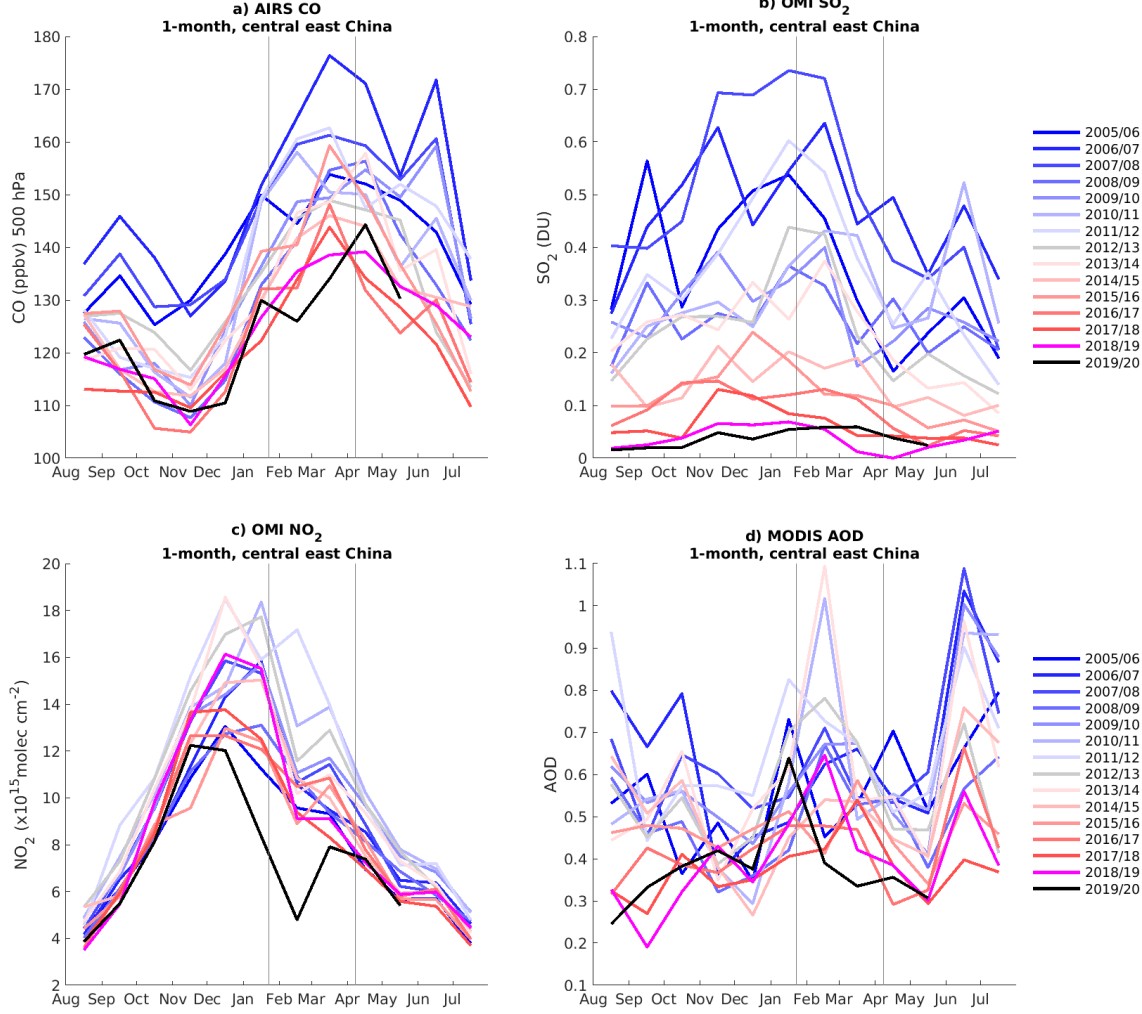

**Figure 3. Monthly mean a) AIRS CO, b) OMI PBL SO₂, c) OMI tropospheric NO₂ and d) MODIS AOD over central east China since 2005. As in Bauwens et al. (2020), each year starts in August to show any departure from the seasonal cycle during the January 23 to April 8 lockdown period, shown by the thin grey vertical lines.**

650





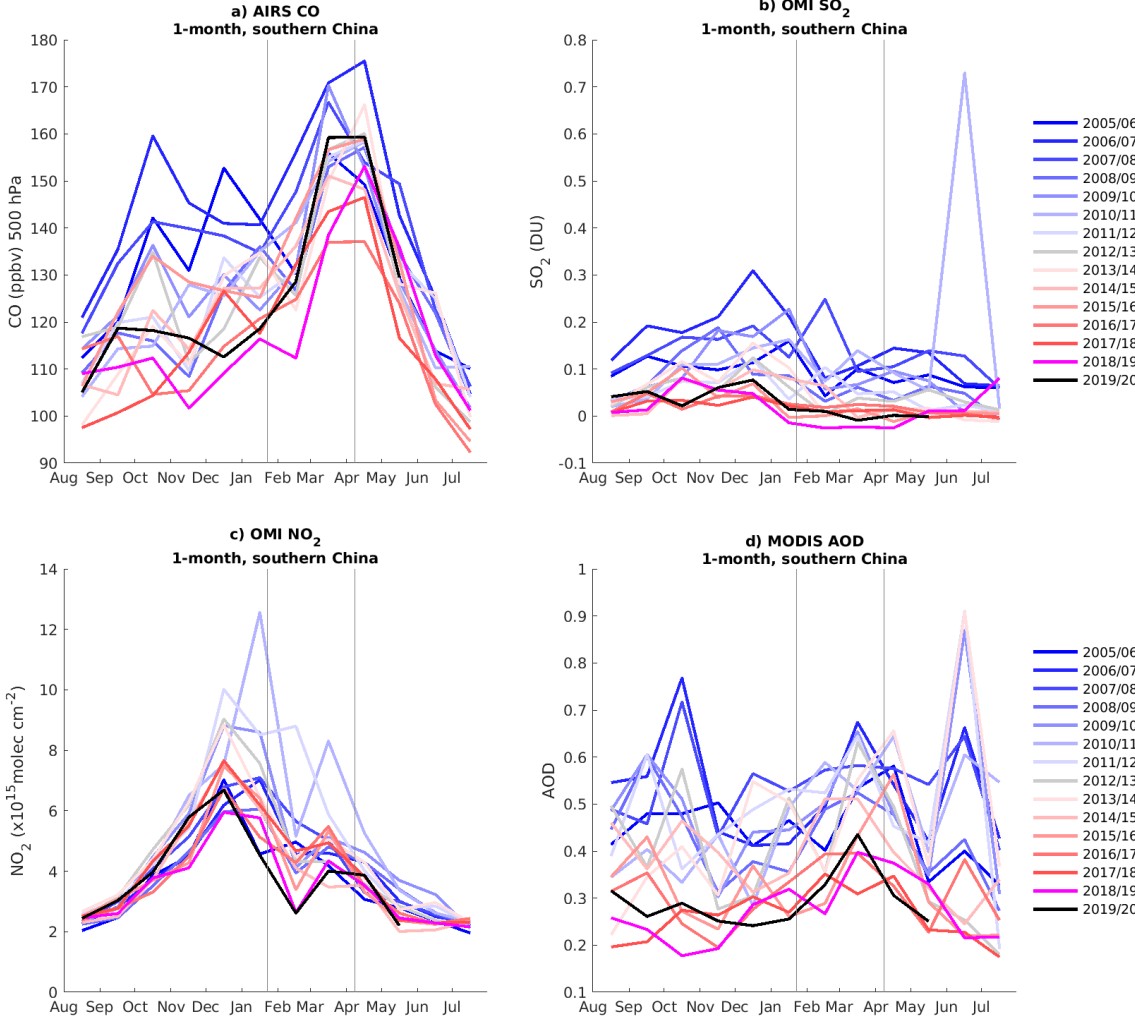

**Figure 4. Same as Figure 3, but for southern China.**

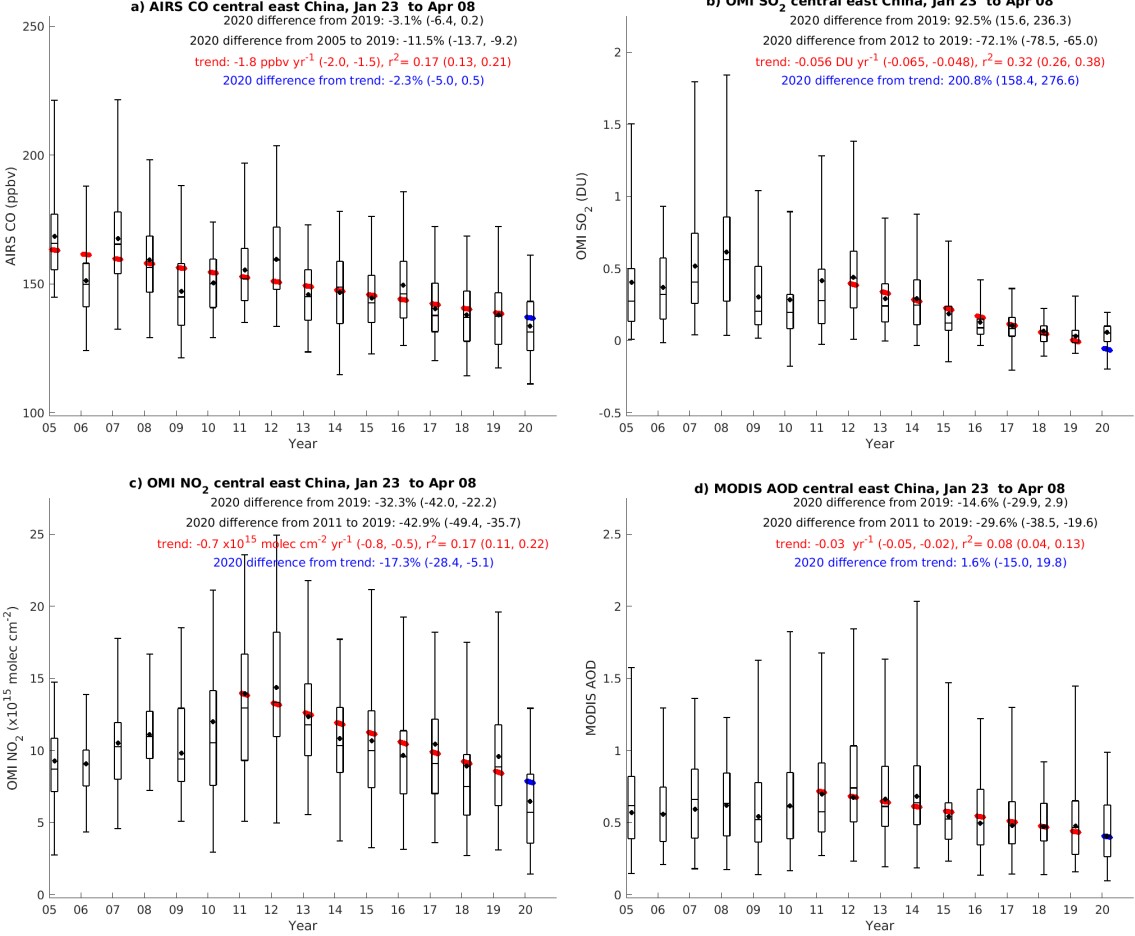

**Figure 5.** January 23-April 8 box plots over central East China for a) AIRS CO, b) OMI PBL SO$_2$, c) OMI tropospheric NO$_2$ and d) Aqua and Terra MODIS AOD from 2005 to 2020. The black box plots show the median, interquartile range and 2.5$^{th}$ and 97.5$^{th}$ percentiles over all daily mean data. For each variable, the estimated trend is plotted in red over the period during which it was strongest and given in the caption with its coefficient of determination (r$^2$). The percentage differences are given between 2020 and 2019, 2020 and the background period, and 2020 and the predicted value from the trend. 95% confidence intervals for each estimate are given in parentheses.



**Figure 6. Dependence of trends (red) and difference between 2020 observations and predicted value (magenta) on detrending start year over central east China for a) AIRS CO, b) OMI PBL SO₂, c) OMI tropospheric NO₂ and d) MODIS AOD. The solid line shows the mean of the estimate for each year and the shading shows the 95% confidence interval.**





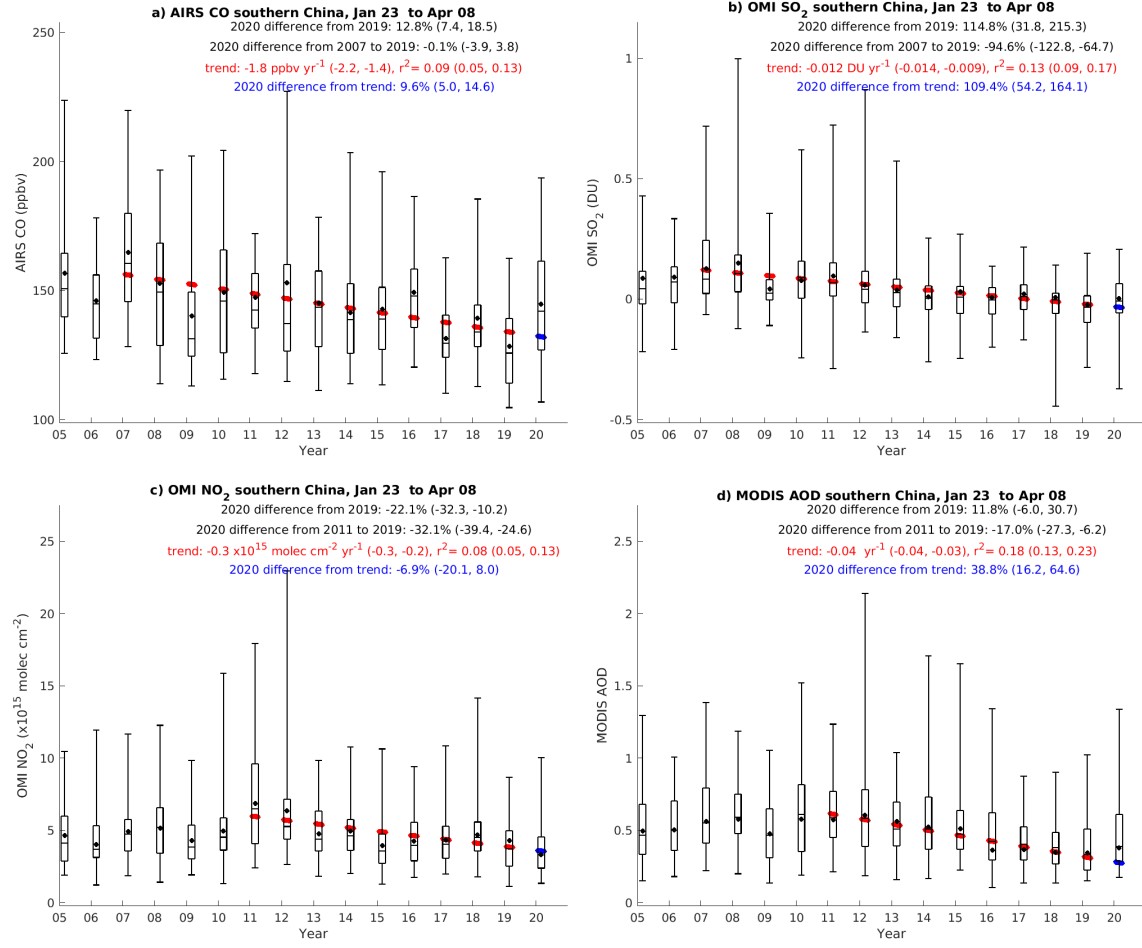

665

**Figure 7. Same as Figure 5 but for southern China.**



**Figure 8. Same as Figure 6, but for southern China.**