# Peer review of "Changes in satellite retrievals of atmospheric composition over eastern China during the 2020 COVID-19 lockdowns"

_Atmospheric Chemistry and Physics, 2020_

## Referee Comment (RC1) · Anonymous Referee #2 · 17 Aug 2020

This article presents an analysis of atmospheric composition changes due to COVID-19-related measures, based on observations from satellites. The methodology adopted in several recent studies for assessing COVID-19 impacts relied on comparisons of pollutants abundances during the lock-downs to the their levels in previous years during the same period of the year. The quantification of lock-down effects is uncertain for several reasons, including natural variability and the long-term trends in anthropogenic emissions. The main point of this paper is that those long-term decreasing trends in emissions explain a great deal of the observed decreases in the abundances of major pollutants like CO, $NO_2$ and aerosols. The paper provides "best estimates" of the COVID-related change in CO, $SO_2$, $NO_2$ columns and aerosol optical depths (AOD)

[Figure]

when taking the long-term trends into account. Accounting for the trends is found to have a huge impact on the inferred Covid-related changes. $NO_2$ is the only compound displaying a significant drop, and it is found to be much lower (17%) than the decrease estimated when ignoring the trend effect.

Major comments:

Although it is of course correct that the long-term trends should ideally be taken into account in order to quantify COVID-related changes, the methodology used here for subtracting the long-term trends is fundamentally flawed. A linear trend is assumed and calculated based on a "background period" defined as the period during which the trend was strongest, based on the $r^2$ value. The flaw is particularly evident in the case of $SO_2$ over East central China: as stated in lines 262-263, the "expected 2020 value" (i.e. the value extrapolated based on a linear trend applied to $SO_2$ data between 2012 and 2019) is negative (-0.06 DU). How can we consider a negative column as an acceptable expected value? This cannot be done. The negative linear trend cannot be extrapolated beyond 2019, because $SO_2$ columns have become so low, and cannot decrease much anymore. It's not just for $SO_2$. For AOD as well, there is a clear flattening of the observed decrease in the later years: as seen by Figure 5, the observed AOD values in 2018-2019 are higher than the linear regression, whereas they are lower than the regression in 2015-2017. This is very clear indication of a flattening. For $NO_2$ as well, the data are lower than the linear fit in 2014-2016, while they are (mostly) higher in 2017-2019. Again, the flattening is evident, and the regression is inappropriate. The choice of the background period based on the $r^2$ value is misguided. Of course, due to natural interannual variability, the $r^2$ of linear fits based on the later years (e.g. 2015-2019) will be higher than the ones adopted here; but it does not imply that they are less appropriate for extrapolation to 2020. Unfortunately, the 95% confidence interval of the trend (Figure 6) is very wide for a start year of 2015: for $NO_2$ and AOD, the range almost spans a zero trend. It would be interesting to see the results for a starting year of 2016. Based on Figure 5, the $NO_2$ and AOD trend

would become close to negligible.

There is another issue of some importance, the definition of the lock-down period. As seen on Figures 3 and 4, the lockdown was much more strict in February than during the rest of the period considered here (January 23 - April 8). In order to separate the effects of the lockdown from other effects, why wouldn't you focus the study on the month of February? Judging from Figure 3, this would make a lot of sense, at least for CO and $NO_2$.

Based on the above, I cannot recommend the paper for publication in ACP. I have serious doubts that the main shortcoming can be remedied. The subtraction of the trend will be always very uncertain, and is very likely a lesser issue than the natural interannual variability related to meteorology.

Minor (language) comments

- lines 66-69: unclear, please rephrase

- l. 72-73: the grammar is incorrect

- l. 133-134: "Systematic errors... have an uncertainty of 20%": rephrase

- l. 170: "higher-quality retrievals": defined how?

- l. 187: "depended on how"

- l. 220: "The main features of the $SO_2$": poor expression, be more specific

- l. 230: "but against a noisy background": unclear

- l. 239: "than in 2019"

- l. 243: "the distribution of daily CO...": this is not what the figures shows. What we see is the evolution of averaged CO etc. abundances over 2005-2020

- l. 247: whisker

---

## Referee Comment (RC2) · Anonymous Referee #1 · 20 Aug 2020

This paper thoroughly discussed the changes in atmospheric composition caused by the blockade policy during the 2020 COVID-19 lockdowns and separated the long-term trend's influencing factors. The author also discussed the dependence of the long-term trend analysis on the starting year and the influence of the starting year to the trend analysis. Questions and suggestions are as follows:

1. AIRS adopted the result of 500 hPa with the highest detection sensitivity in CO products. The data quality of this layer is reliable. However, for CO, whether the information in this layer comes more from the impact of transmission than local emission interference? Is there any information about a layer > 500 hPa with the detection sensitivity

[Figure]

lower than 500 hPa? Can this result compare to the results in this paper?

2. 4b) shows the SO2 distributions. Was the peak of SO2 caused by the high value of a province in South China or the overall high values? Compared to FIG. 4b) and FIG. 4A), the high values are more prominent and significantly different from the other years. What are the reasons?

3. Figure 3 and Figure 4 are not clear. Both figures show monthly changes from 2005 to 2020. However, this paper focuses on the comparison between January 23 and April 8, and the same period in history. Is it better to add a bar chart only focuses on this period?

4. Inline 291, the author stated, "these differences were not consistently significant when daily values were calculated from the median of individual retrievals." Does it possible that the high noise biased the result. The observed data also confirmed this because there was no obvious increased SO2 level during the epidemic period in the Wuhan area. For SO2 with even more considerable uncertainty in atmospheric detection, the consistent result of median value, and the ground-based observation, does it possible that the quality of Omi's satellite SO2 data affects the analysis results? Moreover, if the author can get a consistent conclusion based on the TROPOMI data?

5. This paper discussed South China and central China separately. What is the relationship between them? Line 377 points out that the decrease in Central China is larger than in South China because of the decrease in the NO2 in Wuhan. When doing a similar analysis, we found that the higher the NO2 concentration is, the greater the reduction is. The decrease in South China is smaller than that in Central China, which may be higher than that in Central China (such as Beijing), which is more sensitive to the lockdowns.

6. Line 410, the change of CO trend is attributed to the fact that biomass combustion sources are not affected by the epidemic situation. The possible main reason could be the CO data of 550 HPA does not reflect the influence of human resource, which is

related to the data resources used in this paper. Therefore, this paper concludes that the epidemic situation does not affect the contribution of biomass combustion sources to CO is limited. The limitations of CO data used in this paper on the conclusion should be pointed out, and the impact of the reduction of anthropogenic emissions caused by blockade on CO also needs further discussion.

7. Through the method described in this paper, the changes in atmospheric composition, which are not caused by the epidemic situation, are removed. Can the paper conclude the impact from the epidemic or the combined impact of meteorological factors and the shutdown caused by the epidemic? If it is the latter, it is important to discuss the contribution of meteorological factors.

8. Page 11, Line 409. "We considered transboundary smoke transport as a possible reason . . ..". According to the higher CO level over the Upper Mekong region in 2020, it can be inferred that the CO in southern China increases by transboundary smoke transport (Fig. 2a), so the relevant meteorological environment should be discussed.

9. I suggest future work should be provided analyses on the interaction and relationship between trace gases such as NO2 and CO, and further innovate the study methods by finding the internal mechanism of air pollution and provide the basis for the air pollution source identification.

[Figure]

[Figure]

**b) OMI SO$_2$**
**1-month, southern China**

Legend:
- 2005/06
- 2006/07
- 2007/08
- 2008/09
- 2009/10
- 2010/11
- 2011/12
- 2012/13
- 2013/14
- 2014/15
- 2015/16
- 2016/17
- 2017/18
- 2018/19
- 2019/20

**Fig. 1.** Fig.4b

---

## Author Comment (AC1) · 5 Oct 2020

We thank the referee for their insightful comments and hope that we have done as thorough a job as is practical. We have looked into the spatial nature of the June 2011 OMI $SO_2$ spikes over southern China from the Nabro eruption and whether the main results for AIRS CO at a lower level differed from those for AIRS CO at 500 hPa where the retrieval is most sensitive. We have also mentioned the predominant flow between the upper Mekong region and southern China during the lockdown period. We hope to examine additional meteorological drivers and process-level trace gas and aerosol interactions in follow-up modeling studies including TROPOMI retrievals, mentioned in the Discussion and Conclusions. Point by point responses are provided below.

This paper thoroughly discussed the changes in atmospheric composition caused by the blockade policy during the 2020 COVID-19 lockdowns and separated the long-term trend's influencing factors. The author also discussed the dependence of the long-term trend analysis on the starting year and the influence of the starting year to the trend analysis. Questions and suggestions are as follows:

1. AIRS adopted the result of 500 hPa with the highest detection sensitivity in CO products. The data quality of this layer is reliable. However, for CO, whether the information in this layer comes more from the impact of transmission than local emission interference? Is there any information about a layer > 500 hPa with the detection sensitivity lower than 500 hPa? Can this result compare to the results in this paper?
CO variability is presumably from a mix of local and remote sources, with lower tropospheric CO dominated by local emissions sources, but we cannot say much more than this in a study of this scope, which does not include any modeling or detailed transport analysis. To the best of our knowledge, the retrieval has too few degrees of freedom to analyze different layers separately. For the sake of comparison, however, we compared the AIRS CO for central east China at 500 hPa (below left, as in the manuscript) to that at 850 hPa (below right). As would be expected, the AIRS CO at 850 hPa is ~40 ppbv higher than at 500 hPa, and with a consequently stronger downward trend. The interannual variability is similar between the two altitudes, and there was no significant effect on the differences between CO in 2020.

[Figure]

[Figure]

Over southern China, results were similar in that CO was ~30 ppbv higher at 850 hPa (below right) than 500 hPa (below left), but with comparable interannual variability. The strongest identified trend at 500 hPa (in terms of $r^2$) was from 2016-2019 (which followed from including 2016 as a trend start year in response to the 2nd reviewer's comments), which led to a greater difference between 2020 and the average over that period, and from the expected value from the trend over that period. At 850 hPa, results were more similar to the original 2007-2019 period considered originally.

At L260, we have mentioned that results for CO at 850 hPa are consient with the primary results at 500 hPa.

[Figure]

2. 4b) shows the SO2 distributions. Was the peak of SO2 caused by the high value of a province in South China or the overall high values? Compared to FIG. 4b) and FIG. 4A), the high values are more prominent and significantly different from the other years. What are the reasons?

As mentioned in the original paper, the large SO2 values over southern China in June 2011 were from the Nabro eruption in Ethiopia. Below, the top panel shows the SO2 for May 22-June 10, the two weeks prior to the Nabro eruption. The bottom panel shows the OMI SO2 for June 11-30, the two weeks after the Nabro eruption. The high June 2011 SO2 in the southern China monthly mean (Fig 4b) is from a combination of higher-than average SO2 across the region and also because of individual pixels with much higher retrieved SO2. We have mentioned this at L238. The uneven nature of field is due to frequent clouds. The SO2 over central east China is also higher, but the individual high pixels are largely absent.

[Figure]

[Figure]

3. Figure 3 and Figure 4 are not clear. Both figures show monthly changes from 2005 to 2020. However, this paper focuses on the comparison between January 23 and April 8, and the same period in history. Is it better to add a bar chart only focuses on this period?
We considered this but our preference is to keep this figure as is, to see how the January 23 to April 8 period fits in any seasonality among the retrieved values, which is touched on near the beginning of section 3.1

4. In line 291, the author stated, "these differences were not consistently significant when daily values were calculated from the median of individual retrievals." Does it possible that the high noise biased the result. The observed data also confirmed this because there was no obvious increased SO2 level during the epidemic period in the Wuhan area. For SO2 with even more considerable uncertainty in atmospheric detection, the consistent result of median value, and the ground-based observation, does it possible that the quality of Omi's satellite SO2 data affects the

analysis results? More- over, if the author can get a consistent conclusion based on the TROPOMI data?

Including TROPOMI data is beyond the scope of this study, which is focused on the effects of the COVID19 shutdown in a longer term-context, but we hope to examine TROPOMI in follow-up, modeling-focused studies. But yes, the difficulty in interpreting the SO2 changes is due to properties of the retrieval; whether or not that is noise, a non-normal distribution of values (which we hope have been addressed by comparing the median to the mean), or something else will be a matter for future research.

5. This paper discussed South China and central China separately. What is the relationship between them? Line 377 points out that the decrease in Central China is larger than in South China because of the decrease in the NO2 in Wuhan. When doing a similar analysis, we found that the higher the NO2 concentration is, the greater the reduction is. The decrease in South China is smaller than that in Central China, which may be higher than that in Central China (such as Beijing), which is more sensitive to the lockdowns.

Yes, we agree with this point, and now mention at L439 that the greater reduction over east Central China is due to a higher NO2 'baseline', to the extent that it exists given the trends.

6. Line 410, the change of CO trend is attributed to the fact that biomass combustion sources are not affected by the epidemic situation. The possible main reason could be the CO data of 550 HPA does not reflect the influence of human resource, which is related to the data resources used in this paper. Therefore, this paper concludes that the epidemic situation does not affect the contribution of biomass combustion sources to CO is limited. The limitations of CO data used in this paper on the conclusion should be pointed out, and the impact of the reduction of anthropogenic emissions caused by blockade on CO also needs further discussion.

This is a good point. We have mentioned retrieval limitations at L439 near the surface as a possible reason why some changes in 2020 were not more pronounced in the presence of constituents at higher-altitudes arriving from more remote sources, which can hopefully be addressed through modeling.

7. Through the method described in this paper, the changes in atmospheric composition, which are not caused by the epidemic situation, are removed. Can the paper conclude the impact from the epidemic or the combined impact of meteorological factors and the shutdown caused by the epidemic? If it is the latter, it is important to discuss the contribution of meteorological factors.

We cannot make any such conclusions, as examination of the meteorological contributions to differences between years was beyond the scope of this study. The need to consider meteorological contributions is mentioned toward the end of the paper at L437 and L451. As with the TROPOMI data, we hope to examine in follow-up modeling focused studies.

8. Page 11, Line 409. "We considered transboundary smoke transport as a possible reason . . ..". According to the higher CO level over the Upper Mekong region in 2020, it can be inferred that the CO in southern China increases by transboundary smoke transport (Fig. 2a), so the relevant meteorological environment should be discussed.

The figure below shows the mean MERRA2 850 hPa wind vectors from January 23-April 8 2020, with a west-southwesterly flow from the Upper Mekong region into southern China. This

is broadly consistent with the 2003-2009 mean westerly December-May 700 hPa flow in Reid et al. (2013, *Atm. Res.*), which we have cited at L427.

[Figure]

MERRA2 850 hPa wind vectors and speed, January 23–April 8, 2020

[Figure]

9. I suggest future work should be provided analyses on the interaction and relationship between trace gases such as NO2 and CO, and further innovate the study methods by finding the internal mechanism of air pollution and provide the basis for the air pollution source identification. We agree. As with the comments on satellite retrieval limitations, TROPOMI data, and meteorological factors, we emphasize om the final paragraph the need and expectation of following up with modeling studies looking at aerosol and trace gas interactions at a process level.

[Figure]

**Fig. 1.** Fig.4b

**Anonymous Referee #2**

We appreciate the referee's concerns, which have been addressed point by point below with selected additional analyses, but also their appreciation that long-term trends need to be taken into account. We would argue that our examination of the sensitivity of the 2020 departures to different background and trend periods was thorough, necessarily focused and done specifically to address the referee's concerns, many of which we shared in conducting the analysis.

Regarding the point about negative expected $SO_2$ not being possible, the reviewer is mistaken. Negative $SO_2$ occurs in the OMI retrieval and needs to be included in any trend or average to not introduce a bias. Regarding the choice of lockdown period, we have replaced our shorter-period sensitivity test with a February-only period. But as with original sensitivity shorter-period test, this is problematic because of the different start times of the New Year's holiday, and the circularity of defining the lockdown periods from the data. The main results therefore still focus on the 'official' lockdown period of January 23-April 8. We have added 2016 as a starting trend and background year, which did not significantly affect the main conclusions.

This article presents an analysis of atmospheric composition changes due to COVID-19-related measures, based on observations from satellites. The methodology adopted in several recent studies for assessing COVID-19 impacts relied on comparisons of pollutants abundances during the lock-downs to the their levels in previous years during the same period of the year. The quantification of lock-down effects is uncertain for several reasons, including natural variability and the long-term trends in anthropogenic emissions. The main point of this paper is that those long-term decreasing trends in emissions explain a great deal of the observed decreases in the abundances of major pollutants like CO, NO2 and aerosols. The paper provides "best estimates" of the COVID-related change in CO, SO2, NO2 columns and aerosol optical depths (AOD) when taking the long-term trends into account. Accounting for the trends is found to have a huge impact on the inferred Covid-related changes. NO2 is the only compound displaying a significant drop, and it is found to be much lower (17%) than the decrease estimated when ignoring the trend effect.

Major comments:
Although it is of course correct that the long-term trends should ideally be taken into account in order to quantify COVID-related changes, the methodology used here for subtracting the long-term trends is fundamentally flawed. A linear trend is assumed and calculated based on a "background period" defined as the period during which the trend was strongest, based on the r2 value. The flaw is particularly evident in the case of SO2 over East central China: as stated in lines 262-263, the "expected 2020 value" (i.e. the value extrapolated based on a linear trend applied to SO2 data between 2012 and 2019) is negative (-0.06 DU). How can we consider a negative column as an acceptable expected value? This cannot be done. The negative linear trend cannot be extrapolated beyond 2019, because SO2 columns have become so low, and cannot decrease much anymore.

It's not just for SO2. For AOD as well, there is a clear flattening of the observed decrease in the later years: as seen by Figure 5, the observed AOD values in 2018-2019 are higher than the linear regression, whereas they are lower than the regression in 2015-2017. This is very clear indication of a flattening. For NO2 as well, the data are lower than the linear fit in 2014-2016, while they are (mostly) higher in 2017-2019. Again, the flattening is evident, and the regression is inappropriate. The choice of the background period based on the r2 value is misguided. Of course, due to natural interannual variability, the r2 of linear fits based on the later years (e.g. 2015-2019) will be higher than the ones adopted here; but it does not imply that they are less appropriate for extrapolation to 2020. Unfortunately, the 95% confidence interval of the trend (Figure 6) is very wide for a start year of 2015: for NO2 and AOD, the range almost spans a zero trend. It would be interesting to see the results for a starting year of 2016. Based on Figure 5, the NO2 and AOD trend would become close to negligible.

The SO2 has indeed decreased during the OMI record, which is pointed out at L62, but it is incorrect to say negative SO2 is not possible. Please see the original Li, Joiner et al. (2013, GRL) retrieval paper. This shows negative values for one-month averages (their Figures 2 and 3) and daily spatial averages over the equatorial Pacific (their Table 1). Please also see Wang and Wang (2020, Atm. Env.), noting that negative values also appear in the long-term seasonal (their Figure 4a) and yearly (their Figure 5a) histograms of OMI SO2 over their North China Plain region. The negative values presumably indicate concentrations below the retrieval's detection limit. The expected negative SO2 in 2020 is therefore not impossible, but rather a function of how the retrieval is done and the significant declines in SO2 over the region.

We disagree that the using the r2 to identify periods with consistent trends is misguided, but rather, as stated at L290, "a reasonable but ad-hoc way of defining a period with consistent increasing or decreasing trends.", which is followed by a sensitivity analysis focused on precisely the issue of different trend and background periods, their uncertainty, and their effect on the results. As suggested, we have added 2016 as a trend/background start year in the revised manuscript, shown below for east central China (Figure 6). For CO, a trend starting in 2016 was in fact more strongly negative than if it starts in 2015, not flatter, because 2016 appeared to be anomalously high. But regardless, the CO in 2020 was not significantly different from expected in either case. The SO2 trend starting in 2016 was slightly flatter than in 2015, but also had no significant effect on the 2020 difference. The NO2 trend starting in was technically not significant in 2015, and this is also the case for 2016. This also had no effect on the significance of the 2020 difference.

[Figure]

Yes, the AOD flattening is indeed evident over central east China, particularly in Figure 6d, where, technically, there was no trend over 2015-2019, which is also the case for 2016-2019. This is not unfortunate, but rather an inevitable result of there being fewer data and a flatter trend during more recent years. But for the background/trend period starting in 2016, the AOD is still not significantly different from what would be expected given the considerable spread in the data.

Overall, determining whether a single year's departure is part of flattening or a temporary departure is somewhat subjective, hence our detailed sensitivity analyses in Figures 6 an 8, and which we have emphasize at L445 in concluding the manuscript. We disagree that the use of a linear trend is inappropriate, but rather a valuable, transparent first-order effort to understand trends in concentrations that experience interannual variability.

There is another issue of some importance, the definition of the lock-down period. As seen on Figures 3 and 4, the lockdown was much more strict in February than during the rest of the period considered here (January 23 - April 8). In order to separate the effects of the lockdown from other effects, why wouldn't you focus the study on the month of February? Judging from Figure 3, this would make a lot of sense, at least for CO and NO2.

We agree that the definition of lockdown period is important, which is why we considered shorter and longer lockdown periods in the original manuscript, the rationale for which was provided at the end of the Introduction. To address the referee's comment, the "1-week later/1-week earlier" lockdown period has been replaced with a February-only lockdown period (L200), but knowing that this is also problematic because of the different New Year's holiday periods each year, and with our preference to not define the lockdown periods from the satellite data.

The sensitivity tests for this start period over central east China are shown below. This had some effect on the CO trends in later years, but not on the significance of 2020 differences from what would be expected. This also affect the SO2 in that the positive 2020 difference from expected was no longer significant, but the same issue of disagreement when the median of individual retrievals was used remained, still limiting our ability to draw conclusions from the data. The trends and 2020 differences for NO2 were consistent with those of a January 23-April 8 lockdown period. The AOD 2020 difference from what would be expected was stronger and technically significant, but still with a very wide confidence interval. This has been mentioned in at L348, and the sensitivity test plots have been added as supplementary Figures S5 and S6. But again, our preference is to focus the main analysis on the January 23$^{rd}$ to April 8$^{th}$ period because it includes all New Years holidays from previous years and to not define the lockdown period from the data, and having originally stated at the end of the paper that other approaches could consider region-specific lockdown measures and timing.

[Figure]

[Figure]

Based on the above, I cannot recommend the paper for publication in ACP. I have serious doubts that the main shortcoming can be remedied. The subtraction of the trend will be always very uncertain, and is very likely a lesser issue than the natural interannual variability related to meteorology.

The importance of meteorology is mentioned near the end of the manuscript at L437 and L450 in the context of follow-up modeling work. Following several of the initial COVID-19 related studies described in the Introduction, we disagree that it needs to be examined in this paper, which is focused on accounting for trends and that in doing so, how different trend and baseline period definitions need to be considered – in part, of course, because of the possible contributions of meteorological in interannual variability in the retrieved quantities. How the meteorology contributes alongside the trends to changes in the retrieved quantities will be an important topic for follow-up work, which was stated in the conclusions.

Minor (language) comments

- lines 66-69: unclear, please rephrase
L66: These two sentences have been edited to hopefully be more clear.

- l. 72-73: the grammar is incorrect
L73: Thank you for catching this. It has been corrected.

- l. 133-134: "Systematic errors... have an uncertainty of 20%": rephrase
L133: this has been rephrased to hopefully be more clear.

- l. 170: "higher-quality retrievals": defined how?
L168: We have specified that these are defined as retrievals with less than 30% cloud fraction and not affected by the row anomaly problem, as specified in the OMNO2 README Document Data Product Version 4.0.

- l. 187: "depended on how"
L188: Thank you for catching this, it has been corrected.

- l. 220: "The main features of the SO2": poor expression, be more specific
L221: This has been rephrased to hopefully be more specific.

- l. 230: "but against a noisy background": unclear
L230: This has been rephrased to hopefully be more clear.

- l. 239: "than in 2019"
L240: Thank you for catching this, it has been corrected.

- l. 243: "the distribution of daily CO..." : this is not what the figures shows. What we see is the evolution of averaged CO etc. abundances over 2005-2020
L246: We have described the plots more specifically using the same language as in the caption of Figure 5, also adding that the black dots are the mean of the daily data during the lockdown period for each year.

- l. 247: whisker
This has been removed from the revised version as part of the rephrasing of this section.

---

## Author Response (AR3)

**Editor Decision: Reconsider after major revision**
09 Aug 2021
Please take into consideration the remaining remarks of the referee, in particular the need to better emphasize the existence and importance of the flattening of the observed decrease of SO2, NO2 and AOD in recent years. One important consequence of this, is that the COVID impact estimate is larger than currently claimed in the manuscript.

We appreciate the editor's point about placing more emphasis on the flattening seen in most retrievals during recent years. The Discussion and Conclusions have been changed accordingly; where appropriate, the emphasis has been placed on the differences between 2020 and background means during the different 'flat' periods. The Abstract reflects these changes. At L486, we now conclude by suggesting that follow-up work with additional years of data will help to determine whether this flattening is permanent. The reviewer discussed using a exponential decay curve for regression model. While not 'requiring' this for the revisions, we have suggested this as something to consider in future work at L488.

Regarding the $SO_2$, we have stated at L292 in the Results that the expected 2020 $SO_2$ extrapolated from recent trends is not particularly meaningful, which is followed by the discussion of the effects of calculating the daily SO2 from the median rather than mean retrieved values over the region. We have concluded at L419 that no $SO_2$ changes could be robustly detected in 2020 in either region.

We hope that these changes have struck the right balance between establishing the need to consider past trends and variability, but that COVID-19 related changes in 2020 occurred against a recent flattening during the preceding years.

(25 Jan 2021) by Michel Van Roozendael
Comments to the Author:
The comments raised by referee 2 are fully pertinent. There is an obvious flattening of the trends in NO2 and SO2 in recent years, such that linear trends evaluated from the last decade (2011-2019) cannot be used for extrapolating reference values in 2020. Please consider the suggestion of using trends evaluated over the restricted time period 2019-2019 as a basis for extrapolation in 2020. In any case, revise the main conclusions so that quantitative estimates of the COVID impact clearly reflect the flattening of the trends.

*We appreciate the considerable patience of the Editor and Referee 2 while making these revisions. For consistency, these required additional analysis and more substantial revisions to the text, which we hope have addressed the concerns about the choice of background and trend period.*

*We have moved away from emphasizing any single starting year as a basis for background averaging or extrapolation, which was based previously on the $R^2$ of the trend. Throughout the paper, we have instead emphasized the sensitivity of 2020 differences to the starting year of the background or detrending period, identifying when and where the 2020 differences were significant, the range of these differences, and with what caveats. This shift was made to avoid*

*the difficulties of choosing any single starting year for the baseline period. For NO₂, for example, choosing the 2016-2019 period as a basis is difficult because it is bookended by two years with higher NO₂; which of 2015, 2016 or 2017 to use as a starting year was not obvious to us and inevitably subjective.*

*This shift can be seen in the revised Results section and summarized in the first five paragraphs of the revised Discussion, with the Abstract revised accordingly. In several cases, we have pointed out flattening for more recent periods, for example NO₂ (L349) and AOD (L360) over central east China since 2016, and SO₂ since 2014 (L633) and AOD for more recent years (L651) over southern China. In Figures 5 and 7, the statistics for what were previously the 'strongest' trend years have been removed. The Supplementary Information now contains Tables S1-S8. Each has the background mean, 2020 difference from background mean, trend information, and expected 2020 differences from the trend, all for starting periods between 2005 and 2018, providing expanded information underlying Figures 6 and 8.*

*We hope these changes have addressed the concerns of the 2ⁿᵈ Referee and Editor.*

**Referee 1:**
 - No comments

**Referee 2:**
Review of the revised manuscript by Field et al. "Changes in satellite retrievals of atmospheric composition over eastern China during the 2020 COVID-19 lockdowns"

I am glad that the authors acknowledge the obvious flattening of the observed decrease of SO2, NO2 and AOD in recent years. Nevertheless, they persist in using their wrong reference period (2011-2019) as basis for their extrapolation to 2020. I agree that they do present an analysis of the impact of different reference periods (Fig.6). But their abstract and conclusions are still centred on results which ignore the flattening of the trend, e.g. "OMI NO2 in 2020 over central east China was (...) only 17% less than what would be expected from trends", which is misleading.

The flattening is not an accident. Among several possible explanations for it, the most straightforward is that the potential for further reduction obviously decreases when the column decreases. This is something that the assumption of a linear decrease simply cannot capture. A better regression model would be an exponential decrease. This is simple to achieve: only perform a linear fit of the logarithm of the column, as for example in the recent study of Diamond and Wood (2020). Note that the linear trend of -0.056 DU yr-1 inferred for SO2 (Figure 5) corresponds to a relative trend of -14% yr-1 in 2011, which increases to -100% yr-1 in 2018 and explodes in 2019. Using such a basis for extrapolation to 2020 is meaningless (I acknowledge that negative columns are common in DOAS retrievals, but if, as the authors suggest, they are likely "below detection limits", then I think they should not be used for any extrapolation). Things are less dramatic for NO2, but qualitatively similar, as the NO2 column decreases by almost a factor of 2 between 2011 and 2019. Interestingly, the NO2 trend calculated from 2016-2019 data is 3 times lower than the trend calculated from 2011-2019. This implies

that even the relative trend has diminished in amplitude over the period. The 2016-2019 period is very likely a much more realistic basis for extrapolation to 2020 than 2011-2019.

In conclusion, I do not require further analysis of the data, but I strongly recommend that the authors do better emphasize the existence and importance of the flattening. One important consequence is that the best estimate of the difference attributed to COVID is larger than is currently claimed in this article.

Reference : Diamond, M. S., and Wood, R., Geophys. Res. Lett., 47, e2020GL088913, https://doi.org/10.1029/2020GL088913, 2020.

---

## Author Response (AR4)

The paper has been accepted as is with no further changes requested.